# The Role of Reproductive Modes in Shaping Genetic Diversity in Polyploids: A Comparative Study of Selfing, Outcrossing, and Apomictic *Paspalum* Species

**DOI:** 10.3390/plants14030476

**Published:** 2025-02-06

**Authors:** A. Verena Reutemann, Mara Schedler, Diego H. Hojsgaard, Elsa A. Brugnoli, Alex L. Zilli, Carlos A. Acuña, Ana I. Honfi, Eric J. Martínez

**Affiliations:** 1Grupo de Genética y Mejoramiento de Especies Forrajeras, Instituto de Botánica del Nordeste (CONICET-UNNE), Facultad de Ciencias Agrarias, Universidad Nacional del Nordeste (FCA-UNNE), Corrientes 3400, Argentina; verena.reutemann@agr.unne.edu.ar (A.V.R.); abrugnoli@agr.unne.edu.ar (E.A.B.); azilli@agr.unne.edu.ar (A.L.Z.); cacuna@agr.unne.edu.ar (C.A.A.); 2Estación Experimental Agropecuaria Montecarlo, Instituto Nacional de Tecnología Agropecuaria (INTA), Posadas 3300, Argentina; schedlermara@gmail.com; 3Leibniz Institute of Plant Genetics and Crop Plant Research (IPK), 06466 Gatersleben, Germany; hojsgaard@ipk-gatersleben.de; 4Programa de Estudios Florísticos y Genética Vegetal, Instituto de Biología Subtropical (CONICET-UNaM), Facultad de Ciencias Exactas, Químicas y Naturales, Universidad Nacional de Misiones (FCEQyN-UNaM), Posadas 3300, Argentina; ana.honfi@unam.edu.ar

**Keywords:** apomixis, mating systems, outcrossing, *Paspalum*, population structure, selfing

## Abstract

Exploring the genetic diversity and reproductive strategies of *Paspalum* species is essential for advancing forage grass improvement. We compared morpho-phenological, molecular, and genotypic variation in five tetraploid *Paspalum* species with contrasting mating systems and reproductive modes. Contrary to previous findings, selfing (*Paspalum regnellii* and *P. urvillei*) and outcrossing (*P. durifolium* and *P. ionanthum*) species exhibited similar phenotypic diversity patterns, with low intrapopulation variability and no morphological differentiation among populations. The apomictic species (*P. intermedium*) exhibited low intrapopulation phenotypic variation but high population differentiation, indicative of genetic drift and local adaptation. Outcrossing species showed greater intrapopulation genotypic variation than selfing species, which displayed a high population structure due to restricted pollen migration. The apomictic species exhibited the lowest intrapopulation molecular diversity, forming uniclonal populations with high interpopulation differentiation, highlighting the fixation of distinct gene pools via apomixis. This is the first report about genetic diversity in populations of sexual allopolyploid species of *Paspalum*. Population structure in these allotetraploid *Paspalum* species is primarily shaped by how reproductive modes, mating systems, and geographic distribution influence gene flow via pollen and seeds. Our findings contribute significantly to the conservation and genetic improvement of forage grasses, particularly for developing cultivars with enhanced adaptability and productivity.

## 1. Introduction

Exploring the population genetic diversity of *Paspalum* L. species is essential for conceiving strategies for germplasm collection, conservation, and advancing genetic improvement and cultivar development in this genus [1,2]. Understanding key traits like ploidy levels and reproductive modes and their role in the population patterns of genetic diversity can assist in the development of such strategies [1].

Most angiosperms reproduce through two main modes: sexual and asexual, through either vegetative reproduction or apomixis [3,4]. Sexual reproduction leads to genetically heterogeneous populations as new genotypes emerge through meiotic recombination and random syngamy [5]. In addition, sexual reproduction avoids mutation accumulation by natural selection of deleterious combinations [6]. Additionally, sexually reproducing plants experience regular genetic exchange between individuals and populations via pollen and seed dispersal, reducing interpopulation differentiation and the potential impact of genetic drift [5].

On the contrary, apomictic plants show reduced genetic variation due to the absence of meiotic segregation and recombination, as demonstrated in numerous studies [7,8]. Thus, apomictic plants produce offspring genetically identical to the mother plant through seeds [7]. There are two types of apomixis: sporophytic and gametophytic, which differ in how the clonal embryos are formed [3,7]. In sporophytic apomixis, also called polyembryony, an embryo originates from the nucellus or the integuments independently of the gametophytic development [3]. In gametophytic apomixis, an unreduced female gametophyte is formed in which the egg cell develops into an embryo without fertilization (parthenogenetically), and the seeds are developed either autonomously (without pollination) or requiring fertilization of the central cell (pseudogamy) for successful endosperm development [7]. Therefore, another expected consequence of apomixis is the formation of genetically uniform (uniclonal) populations that become differentiated through stochastic and/or adaptive events [9,10,11,12]. This usually results in multiple origins and founding events, leading to high population differentiation through the fixation of divergent genotypes [13,14,15,16]. Apomixis has long been seen as an unparalleled natural tool to exploit plant breeding, with a potential influence on global farming systems [1,17].

In addition to reproductive modes, mating systems also play a pivotal role in shaping the genetic structure of plant populations [18]. Selfing tends to evolve when the advantages of transmission and reproductive assurance outweigh the disadvantages associated with inbreeding depression and reduced fitness caused by the increased homozygosity of deleterious alleles [19]. However, selfing is expected to evolve at the cost of reduced genetic variation [18,20,21]. Several studies have shown that selfing species exhibit lower intrapopulation variation but increased interpopulation differentiation than outcrossers due to genetic drift and limited gene flow [21,22,23,24,25,26,27,28]. More recently, comparative population genetic studies of related plant species have confirmed this pattern [29,30].

Beyond reproduction modes and mating systems, environmental and ecological factors also influence the spatial genetic structure of plant species [31,32,33]. Genetic diversity results from a complex interplay of intra- and interpopulation dynamics, as well as climatic, historical, and geological factors [5,34]. Understanding these dynamics is crucial for developing local conservation strategies [35] and promoting sustainable agroecosystems [36].

*Paspalum* is one of the few genera in which sexual and asexual reproductive modes have been characterized side by side for more than 70 years [2]. Sexuality in *Paspalum* species is frequent and is characterized by the double fertilization of a reduced embryo sac of the *Polygonum* type [37]. Over 330 species have been described for this genus [38]; however, only a few of them have been studied at a population level. Cytoembryological and cytological analyses of 72 *Paspalum* species showed that 68% of them had some potential for apomixis, 73.6% showed self-fertility, 43.1% were self-sterile, and 75% were polyploids [39]. Nevertheless, these species only represent 21.8% of the total number of species in the genus, and most studies involve singular plants rather than populations. The study of reproduction modes and mating systems in *Paspalum* species offers valuable insights into the genetic mechanisms underlying population genetic structure, as observed in previous works on this genus [16,30,40,41,42,43]. This knowledge is essential for the efficient use of these tropical grasses in breeding programs and conservation plans [1]. Therefore, we selected five *Paspalum* species with contrasting reproductive modes and mating systems, commonly found in the natural grasslands of northeastern Argentina.

*Paspalum durifolium* Mez and *P. ionanthum* Chase are allotetraploids (2*n* = 4*x* = 40) that form complexes with higher ploidy numbers [44,45,46,47]. These tetraploids are sexual and predominantly outcrossers due to a gametophytic self-incompatibility system (GSI) [48]. Although some tetraploids showed apomictic embryo sacs, no apomictic seeds have been reported [48]. *Paspalum regnellii* Mez and *P. urvillei* Steud. are allotetraploid species (2*n* = 4*x* = 40) [49,50,51]. Both species behave mainly as selfers, but varying degrees of cross-pollination have been reported [48]. Finally, *P. intermedium* Munro ex Morong is an agamic complex with self-sterile sexual diploids (2*n* = 2*x* = 20) and self-fertile facultative–apomictic autotetraploids (2*n* = 4*x* = 40) [44,52]. Species like *P. ionanthum*, *P. regnellii*, and *P. urvillei* hold promise for domestication or genetic improvement as forage species due to their desirable agronomic traits. In contrast, *P. durifolium* and *P. intermedium* typically form large tussocks, which are less suitable for grazing and have lower forage value but have potential as ornamental crops.

In this study, we addressed the following questions: (1) Do populations of self-fertile *Paspalum regnellii* and *P. urvillei* exhibit lower genetic diversity than self-sterile populations *P. durifolium* and *P. ionanthum*? (2) Is genetic diversity lower in apomictic *P. intermedium* compared to self-fertile and self-sterile *Paspalum* species? (3) Are the differences in the genetic structure among *Paspalum* populations related to their mode of reproduction or mating system?

## 2. Results

### 2.1. Phenotypic Variation Within Populations

We described the variation within each population using the mean and standard deviation (Table 1). The coefficient of variation for morphological traits indicated low variation within populations for most traits. Exceptions included internode length (*NL*) in the PIN7 population of *P. intermedium*, as well as the PD1 and PD4 populations of *P. durifolium* and the vegetative period (*VP*) in PU2, PU3, PU4, and PU5 populations of *P. urvillei* (Appendix A).

### 2.2. Phenotypic Variation Among Populations

We observed a high correlation between vegetative and flowering periods (*VP-FP*) in *P. intermedium*, *P. durifolium*, *P. ionanthum*, and *P. urvillei* (>0.9; Appendix A). Additionally, we found a high correlation between plant height and the length of the flowering culms with inflorescences (*H-VL*) in *P. intermedium* and *P. ionanthum*, as well as between inflorescence length and the basal raceme length (*IL-BRL*) in *P. regnellii* (>0.9; Appendix A). Therefore, in the MANOVA analysis, we removed *VP* for *P. intermedium*, *P. durifolium*, *P. ionanthum*, and *P. urvillei* and *VL* for *P. intermedium* and *P. ionanthum*, and *BRL* for *P. regnellii*.

*Paspalum intermedium* showed significant differentiation among populations for all traits (Pillai’s test, *F* = 21.08, num. D.f. = 52, den. D.f. = 288, *p* < 0.001; Appendix A). *Paspalum durifolium* exhibited differentiation among populations (Pillai’s test, *F* = 22.83, num. D.f. = 56, den. D.f. = 284, *p* < 0.001) only in *FP* and *LL* (Appendix A). Pairwise comparisons revealed that the population PD1 differed from PD2, PD4, and PD5 in *FP*, while PD4 was differentiated from PD1, PD2, and PD5 in *LL* (Appendix A). *Paspalum ionanthum* exhibited differentiation among populations (Pillai’s test, *F*= 57.88, num. D.f. = 52, den. D.f. = 328, *p* < 0.001) in *FP*, *H*, *LL*, *NL*, *SL*, *ShL*, and *SW* (Appendix A). *Paspalum regnellii* showed differentiation among populations (Pillai’s test, *F* = 42.61, num. D.f. = 48, den. D.f. = 336, *p* < 0.001) in *EBR*, *FP*, *H*, *Hw*, *ShL*, *VL*, and *VP* (Appendix A). *Paspalum urvillei* showed differentiation among populations (Pillai’s test, *F*= 40.69, num. D.f. = 52, den. D.f. = 296, *p* < 0.001) in *FP*, *H*, *Hw*, *LL*, *LW*, and *SL* (Appendix A). Differentiation among populations for each trait and species is detailed in Appendix A.

We observed an intermediate correlation between phenotypic variation and geographic distribution in apomictic *P. intermedium* (Mantel statistic *R* = 0.41, *p* = 0.001). A low correlation between phenotypic variation and geographical distribution was found in *P. ionanthum* (Mantel statistic *R* = 0.12, *p* = 0.015), *P. regnellii* (Mantel statistic *R* = 0.14, *p* = 0.001), and *P. urvillei* (Mantel statistic *R* = 0.17, *p* = 0.001), but no significant correlation was observed in *P. durifolium* (Mantel statistic *R* = 0.04, *p* = 0.141).

The UPGMA analysis showed that PIN7 of *P. intermedium* clustered separately (Figure 1E). Population PIN4 and PIN9 formed two clusters each, and PIN8 formed four groups; one of them grouped with PIN2 and another with PIN9 (Figure 1E). The k-means clustering (K = 1–10) indicated eight well-differentiated groups in the apomictic species *P. intermedium* (optimal number of clusters = 8, Figure 1E). In contrast, the four sexual species did not show clear clustering among individuals from the same population (Figure 1A–D). Additionally, the k-means analysis indicated no well-differentiated groups in the sexual species (optimal number of clusters = 1, Figure 1A–D).

The PCA for *P. intermedium* explained 72.9% of the pheno-morphological variability within the first two axes, with a significant contribution of *FP*, *Hw*, *NR*, *IL*, *H*, and *SL* to these axes (Appendix A). Along axis 1, population PIN4 was separated towards the positive pole, displaying high values for most of the traits, while populations PIN2 and PIN7 were separated toward the negative pole, showing high values of *FP* (Figure 2A). Over axis 2, populations PIN8 and PIN9 were separated from the other three populations, by showing low values of *BRL*, *EBR*, *SL*, and *SW* (Figure 2A).

The PCA for *P. durifolium* explained 40.4% of the distribution of the variability within the first two axes and showed that *NL*, *VL*, and *H* contributed the most to these axes (Appendix A). The PCA for *P. ionanthum* explained 39.8% of the distribution of the variability within the first two axes and showed that *BRL*, *IL*, *VP*, and *FP* contributed the most to these axes (Appendix A). The PCA for *P. regnellii* explained 37.3% of the distribution of the variability within the first two axes and showed that *VL*, *H*, and *IL* contributed the most to these axes (Appendix A). The PCA for *P. urvillei* explained 38.4% of the distribution of the variability within the first two axes and showed that *IL*, *H*, and *NL* contributed the most to these axes (Appendix A). As observed in the UPGMA and k-means, there was no distinctive separation among populations of the sexual species (Figure 2B–E). Only *P. ionanthum* showed a separation among populations PI1 and PI5 over the axis Dim2 (Figure 2C).

### 2.3. Molecular and Genotypic Diversity Within Species

A total of 124 ISSR fragments for *P. durifolium*, 139 for *P. ionanthum*, 122 for *P. regnellii*, 102 for *P. urvillei*, and 90 for *P. intermedium* were analyzed.

The apomictic populations of *P. intermedium* displayed the lowest values of molecular and genotypic diversity, with each population represented by only one genotype (Table 2). Populations of self-incompatible *P. durifolium* and *P. ionanthum* exhibited high polymorphism and high genotypic diversity indexes (Table 2). One private fragment (*PB*) was found in population PI4 of *P. ionanthum* (Table 2). In contrast, molecular diversity in self-compatible *P. regnellii* and *P. urvillei* populations was low, especially regarding polymorphism. Genotypic variability was higher in populations of *P. urvillei* than in *P. regnellii*, which showed low numbers of effective genotypes and, consequently, low genotypic diversity values (Table 2). Although unexpected for a sexual species, individuals in *P. regnellii* sharing the same genotype exhibited differences at the morpho-phenological level (Figure 2C).

### 2.4. Genetic Differentiation and Gene Flow

Genetic differentiation among populations was the highest in *P. intermedium*, as each population was represented by a unique genotype (Table 3). Differentiation was low in self-incompatible *P. durifolium* and intermediate in *P. ionanthum* (Table 3). A high population structure was observed in self-compatible *P. regnellii* and *P. urvillei* (Table 3). The pairwise F_ST_ values indicated intermediate differentiation for population PIN4 of *P. intermedium* with PIN2, PIN7, and PIN8 (pairwise F_ST_ > 0.12, Table 4). Populations of sexual species showed very low genetic differentiation (pairwise F_ST_ < 0.05, Table 4).

The Mantel test demonstrated that genetic variation was geographically correlated in populations of *P. intermedium* (R^2^ = 0.47, *p* = 0.001), and the isolation by distance test (IBD) inferred eight distance clusters, with six of them showing significant isolation (*p* < 0.05, Appendix A). Genetic variation was correlated with the geographical distribution of populations in self-incompatible *P. ionanthum* (R^2^ = 0.36, *p* = 0.001) and *P. durifolium* (R^2^ = 0.13, *p* = 0.001), and self-compatible *P. regnellii* (R^2^ = 0.34, *p* = 0.001) and *P. urvillei* (R^2^ = 0.25, *p* = 0.001). The IBD identified seven clusters in *P. durifolium*, with two of them isolated by distance (*p* < 0.05, Appendix A), six clusters in *P. ionanthum* with four of them isolated (*p* < 0.05, Appendix A), seven clusters in *P. regnellii* with four of them isolated (*p* < 0.05, Appendix A), and eight clusters in *P. urvillei* of which two clusters were isolated (*p* < 0.05, Appendix A).

### 2.5. Population Structure and Cluster Analysis

Five clusters with a high bootstrap value (>90%) were identified in the unrooted UPGMA dendrogram for *P. intermedium*, each corresponding to a unique population (Figure 3A). The k-means algorithm and structure analysis corroborated these five clusters (Appendix A) with no genetic admixture among them (Figure 3B,C). This high population differentiation was confirmed by the DAPC, showing a clear separation among populations of *P. intermedium* (Figure 3D).

For *P. durifolium*, a single high bootstrap cluster (100%) was recovered, with no preferential clustering among populations (Figure 4B). The k-means analysis also detected a single cluster for *P. durifolium* (Appendix A). The structure analysis showed three distinctive clusters for *P. durifolium* (Figure 4A,C). Populations PD1 and PD2 showed low levels of admixture, while the other three populations showed intermediate levels of admixture and were composed mostly of one genetic cluster (Figure 4C). According to the DAPC, populations PD1 and PD5 are genetically different over PC1, and populations PD2-3-4 are genetically similar (Figure 4D).

There was only one cluster with a high bootstrap (100%) for *P. ionanthum*, but individuals showed preferential clustering within their original population (Figure 5B). The k-means analysis identified four clusters in *P. ionanthum* (Appendix A). The structure analysis in *P. ionanthum* also showed four clusters (Figure 5A,C) with low levels of admixture except for PI3, which showed intermediate levels of admixture in some individuals (Figure 5A,C). Populations PI1 and PI5 are genetically different according to the DAPC, which could correspond with their geographical separation as estimated by the Mantel test (Figure 5D). However geographical distances do not explain the high differentiation among population PI2 and PI3-PI4.

The UPGMA clustering identified only one group with a high bootstrap (100%) for *P. regnellii*, but some individuals from the same local populations were clustered together (Figure 6B), and the k-means recovered three clusters in this species (Appendix A). The structure analysis of *P. regnellii* showed four clusters, which had high levels of admixture in populations PR1 and PR2, an intermediate proportion of admixture in populations PR3 and PR4, and a low proportion of admixture in PR5 (Figure 6A,C). The southern population PR5 showed a clear differentiation from the other four populations over PC1 in the DAPC (Figure 6D).

As observed in the other three sexual species, only one group with a high bootstrap (100%) was observed for *P. urvillei*, with some individuals from the same population clustering together (Figure 7B), and the k-means analysis detected two clusters in this species (Appendix A). However, the Bayesian analysis identified four genetic clusters, with PU4 showing a low proportion of admixture, PU2-3-4 showing intermediate admixture coefficients, and PU5 showing a high admixture proportion in some individuals (Figure 7A,C). In the map, western populations (PU4 and PU5) showed a clear differentiation from the eastern populations (Figure 7C), and the DAPC showed the separation of PU5 from the other four populations over PC1 (Figure 7D).

## 3. Discussion

The reproductive mode is one of the main factors determining the genetic structure of populations [18,53]. Whether a species is predominantly a selfer, outcrosser, or apomictic significantly influences how genetic diversity is distributed within and between populations. In self-incompatible sexual species, genetic diversity is typically high within populations and low between populations [22,23,54]. In contrast, self-compatible sexual and apomictic species usually exhibit the highest genetic diversity between populations [22,23,55]. Thus, understanding the reproductive system is essential for interpreting the genetic diversity, structure, and evolutionary dynamics in allopolyploid species.

### 3.1. Opposite Mating Systems Led to Similar Patterns of Phenotypic Variation but Differed in Apomictic Species

According to Hamrick and Godt [23], obligate allogamous species showed high intrapopulation variability and low variability between populations. By contrast, species that are self-compatible or reproduce via apomixis exhibit low intrapopulation variability and high interpopulation variability [23,55]. Our results showed that the four sexual *Paspalum* species included in this study showed similar patterns in the distribution of phenotypic variation, unlike the general consensus. Both outcrossing (*P. durifolium* and *P. ionanthum*) and selfing (*P. regnellii* and *P. urvillei*) species had low intrapopulation variation and little to no differentiation among populations. This general low intrapopulation variability in vegetative and reproductive traits is a consequence of cross-pollination in obligated allogamous species [56]. In allogamous populations of *Paspalum indecorum*, cross-pollination reduced interpopulation differentiation and normalized phenotypes within populations [30]. In selfing species, greater differentiation among populations was expected, as usually observed in diploid species [57]. For instance, selfing increases the rate of homozygous genotypes, leading to contrasting phenotypes within populations [23]. These contrasting phenotypes widen the range of phenotypic intrapopulation variation and reduce interpopulation differentiation, as observed in populations of *Paspalum pumilum* [30].

Fragmentation and reduction in population size due to anthropogenic activities have likely contributed to the observed reduction in genetic diversity [58], especially in self-compatible species. Thus, low intrapopulation variation in selfing populations of *P. regnellii* and *P. urvillei* may reflect a recent bottleneck, as they were collected from disturbed environments, such as roadsides or areas impacted by pine plantations. The low observed levels of phenotypic variation may also reflect a low representativeness of the sample sizes analyzed for each population (N = 14–20) and the constraints in detecting significant differentiation among individuals or populations [59]. However, in *P. urvillei*, high variation in the vegetative period (VP) was observed within populations, which is important for late flowering line selection, a trait usually linked to higher yield and nutritional value in C4 forage plants [60,61]. In wind-pollinated species, variation in flowering time may reflect adaptation to environmental differences, such as photoperiod, temperature, and water availability [62,63]. Hence, this variation could reflect differences in the adaptive responses of plants to their natural environments, as found in common garden experiments [62,64].

### 3.2. Local Adaptation and Phenotypic Differentiation Among Populations

The analysis of interpopulation variance revealed significant differences between populations in several traits. Populations of the outcrosser *P. ionanthum* and selfers *P. regnellii* and *P. urvillei* showed low correlations between phenotypic and geographic distances, suggesting that phenotypic variation does not reflect local adaptation to environmental conditions [57,65,66,67,68]. Moreover, phenotypic analyses (PCAs) showed overlapping phenotypes among populations and no differentiation in both outcrossing and selfing species. This finding is consistent with the UPGMA analysis, which identified only one phenotypic group for each species, suggesting that wind cross-pollination in outcrossing species (*P. durifolium* and *P. ionanthum*) maintains populations as cohesive phenotypic units [57,69,70]. In selfing species, increased homozygosity can lead to contrasting phenotypes, thereby widening the range of phenotypic variation within populations and decreasing differentiation between populations [30]. The limited geographic distribution of the collected *P. regnellii* populations in Misiones may also explain the low differentiation, as reduced environmental heterogeneity and potential seed migration between locations could homogenize phenotypic traits [71,72]. In *P. urvillei*, the observed phenotypic homogeneity among populations could reflect an increased rate of cross-pollination, as previously observed in these populations [48]. In species with mixed mating systems, in which reproduction occurs via both selfing and outcrossing [73], pollen dispersal among populations can also contribute to low differentiation.

### 3.3. Apomixis Leads to Fixation of Adapted Phenotypes

Apomixis could lead to the fixation of genotypes that are well adapted to their ecological niches [74]. Combined with genetic drift or natural selection, apomixis leads to clonal populations with low evolutionary flexibility but high local specialization [56]. This characteristic of apomictic species is often reflected in low phenotypic variation within populations and high differentiation between populations [55]. Apomictic populations of *P. intermedium* were morphologically uniform within populations but displayed significant differentiation among populations across all phenotypic traits, clustering into eight distinct morphological groups. In contrast to sexual species, apomictic *P. intermedium* showed narrow phenotypic variation, with only one population showing high internode length (NL) variation. Similar patterns of low variation in quantitative traits have been reported in other apomictic species of *Paspalum* [16,75,76]. The high interpopulation differentiation in *P. intermedium* indicates that each population retains a portion of the phenotypic variation fixed by apomixis, and the significant geographic correlation suggests adaptation to local environments [43]. New adaptive or neutral mutations that arise during rare recombination events (i.e., residual sexuality) may be maintained within apomictic populations, leading to the propagation of different clones throughout the distribution range [3,6,77,78,79]. Morpho-phenological analyses like these highlight the importance of doing prior assessments of multiple populations. These studies help in designing germplasm conservation programs that preserve all phenotypic clusters, besides helping in the selection of contrasting phenotypes for future breeding programs.

### 3.4. Contrasting Patterns of Intrapopulation Variation Among Selfing and Outcrossing Species

Selfing species typically have smaller effective population sizes and lower recombination rates, leading to reduced genetic diversity, increased linkage disequilibrium, and higher homozygosity compared with outcrossing species [18,21]. Numerous studies have shown that selfing species have lower genetic diversity and heterozygosity at the population level compared to outcrossing species [25,26,28,30,72,80].

Our results for outcrossing species (*P. durifolium* and *P. ionanthum*) showed high molecular and genotypic variation within populations, whereas selfing species (*P. regnellii* and *P. urvillei*) displayed lower molecular diversity and genotypic variation. In *P. regnellii*, almost all populations exhibited low numbers of effective genotypes (G), decreasing genotypic diversity. Similar values have been observed in selfing populations of *Hordeum spontaneum* [81] and *Lupinus mutabilis* [82]. In the diploid self-compatible *Paspalum pumilum*, genotype-sharing individuals were attributed to either ramets or shared allelic conditions in the analyzed loci [30]. Because ISSRs are dominant markers, we cannot assume the allelic condition in the individuals with the same multilocus genotype, but this is more plausible because our experimental design avoided collecting ramets. Differences in molecular and genotypic diversity between *P. regnellii* and *P. urvillei* could be due to varying rates of allogamy between these species [48]. As observed in other autogamous species, even low rates of sporadic allogamy can lead to high intrapopulation variation and polymorphism [82,83,84]. A mixed pollination system can promote greater diversity compared to strict selfing but is generally lower than in outcrossing species [84,85]. The geographic distribution of populations could also play a role in the intrapopulation genetic diversity. Widespread species usually show significantly higher levels of genetic and genotypic diversity than geographically restricted species [18,22]. Populations of *P. urvillei* were collected in five different Argentinian provinces (see Figure 7), while populations of *P. regnellii* were restricted to Misiones. The wide geographical distribution of the collected populations combined with the low rates of allogamy-originated seeds of *P. urvillei* could contribute to the differentiation among these selfing species regarding intrapopulation genetic diversity.

### 3.5. Differences in Genetic Structure Among Mating Systems

Pollen migration in selfing plants is rare; thus, the lack of gene flow among populations increases genetic differentiation among populations due to genetic drift and fixation of different loci within genetically isolated populations [86,87,88,89]. Therefore, selfing species, compared to outcrossing ones, usually show higher interpopulation differentiation, mainly due to the limited gene flow via pollen [86,89]. In our study, selfing species showed higher values of Rho_ST_ than outcrossing species, but *P. ionanthum* showed a significantly high Rho_ST_, even though it is a strictly allogamous species (gametophytic self-incompatibility system) [48]. The genetic diversity observed with molecular markers was geographically correlated in all sexual species, and some of the identified clusters were isolated by distance. Compared with outcrossing species, selfing species always tend to show a stronger spatial genetic structure, due to a lack of gene flow via pollen [29,89]; however, this was not evidenced in the comparison between selfing and outcrossing *Paspalum* species. This could indicate strong isolation among populations intensified by local adaptation [29,31,90]. Topography and climate differences could explain the absence of pollen movement and restricted seed dispersal in these *Paspalum* species. Topographic barriers, such as the Paraná River, Iberá Wetlands, land-use–land-cover changes, and climatic differences, like precipitation, could further contribute to genetic differentiation among populations as observed in our DAPC and Bayesian analysis of genetic clusters.

### 3.6. Population Structure in Apomictic Species Showed No Admixture

The evolutionary success of an agamic species depends on the possibility of eventually acquiring genetic variability to adapt to long-term environmental fluctuations. However, genetic uniformity is expected within populations of strictly asexual species [10,11,12]. In *P. intermedium*, low molecular and genotypic variation was observed within populations, with all populations being essentially uniclonal. These findings align with observations in other apomictic *Paspalum* species, where intrapopulation diversity was minimal [16,42,43]. Low genotypic diversity is typical of apomictic taxa, as the absence of meiotic recombination restricts the generation of new genetic variants. The limited intrapopulation diversity in *P. intermedium* likely reflects the absence of gene flow and fixation of independent mutations within populations [43]. Some populations exhibited private genetic bands, which could be due to random mutations fixed by genetic drift [8,91,92]. Furthermore, independent genotypes could prove advantageous in each local population, consequently fixing a portion of the whole genetic variation [89,93,94,95,96], as suggested by the uniclonal populations of *P. intermedium*. Additionally, habitats with contrasting environments may favor the fixation of locally adapted genotypes, promoting high interpopulation diversity [13,14,43,97,98]. Population structure analyses revealed that apomictic populations exhibited higher differentiation between populations than sexual species, with almost all genetic variation found among populations (85%). These findings align with previous studies on other apomictic taxa, such as *Limonium dufourii* [99] and *Ranunculus carpaticola* [95]. High levels of clonal clustering are common in apomictic species due to limited dispersal ability [11,13,100,101], as was observed by the results of our cluster analysis, showing that individuals within local populations of *P. intermedium* clustered together. These results agree with the observation by Karunarathne and Hojsgaard [43] of recurrent autopolyploidization events in genetically distinct diploid populations, consequently fixing dissimilar portions of the diploid’s gene pool. Moreover, there is an apparent lack of seed migration among populations of this apomictic species, rendering clusters without admixture unlike sexual species, as observed in the Bayesian analysis and isolation-by-distance (IBD) tests. Geographical barriers, like the Parana River or Iberá Wetlands, could enhance even more the differentiation among populations, hindering seed exchange among populations [31,33,34,69].

## 4. Materials and Methods

### 4.1. Plant Material

Twenty-five populations of five tetraploid species were collected from northeastern Argentina (Table 5). Ploidy levels and reproductive modes were previously described in Schedler et al. [48]. Sampling was conducted from November to March during the years 2013 to 2015. Natural distribution of each species according to Zuloaga and Morrone [102] was considered for population sampling. The sampling sites covered the distribution range of these species in northeastern Argentina. Within the natural area of each species, only populations with a linear size ≥ 200 m that were monoploid and tetraploid were considered for this work (see Schedler et al. [48]). We selected populations that were at least 50 km apart. A uniform sampling strategy was employed, ensuring even representation within each population by maintaining a distance of 10 m between consecutive individuals. Five populations, with approximately 20 plants per population, were collected for each species. Moreover, habitat type was recorded for each species as either an undisturbed habitat (monotypic or transition grassland) or disturbed habitat (when the species was on road edges or forestation margins) (see Table 5). Only plants that survived two years post-transplant in the field were used in the phenotypic and genetic analysis (N, Table 5). Rhizome cuttings of each plant were cultivated in pots within a greenhouse. Once rooted, the plants were transferred to the experimental field of FCA-UNNE in Corrientes, Argentina. Herbarium vouchers for each sampled population were collected and deposited at the CTES and MNES Herbaria.

### 4.2. Morpho Phenological Traits

Morphological variation was assessed using 13 phenotypic traits, while phenological variation was estimated using two traits. Measurements were made during the flowering peak of each species, and all populations were cultivated under the same environment. The traits measured included plant height with (H; cm) and without flowering culms (Hw; cm), 2nd leaf blade length (LL; cm) and width (LW; cm), 2nd leaf sheath length (ShL; cm), 2nd internode length (NL; cm), inflorescence length (IL; cm), basal raceme length (BRL; cm), number of spikelets in the basal raceme (EBR), raceme number (NR), flowering culms’ length with inflorescences (VL; cm), and spikelet length (SL; mm) and width (SW; mm). The phenological traits measured were the extension of the vegetative period (VP; days) and the flowering period (FP; days).

### 4.3. Statistical Analyses of Morpho Phenological Traits

To assess the variation within populations, we calculated the mean, standard deviation (s.d.), and coefficient of variation (C.V.) for each trait within each population (Table 2; Appendix A) using the rstatix 0.7.2 package in R [103]. We established an upper limit of 30% C.V. to define homogeneity within populations (indicating low dispersion around the mean). Values exceeding 30% were considered indicative of heterogeneity among individuals within the same population (Appendix A).

We evaluated variability within and between populations using MANOVA with Pillai’s statistic, testing univariate and multivariate assumptions for a normal distribution using the rstatix 0.7.2 package in R. Non-normal variables were transformed using a logarithmic function. Variables that remained non-normal after transformation were excluded from the MANOVA. A Pearson’s correlation test was conducted to assess the relationships between variables, and only those with a pairwise Pearson coefficient >−0.9 and <0.9 were considered (Appendix A), as MANOVA is sensitive to correlated variables over 0.9 or −0.9. We used a Kruskal–Wallis test to identify which morpho-phenological traits showed significant differences among populations (Appendix A). A post hoc test using Games–Howell’s statistic with Tukey’s p-adjustment was performed to identify specific population differences (Appendix A).

To identify clustering patterns, we constructed a phenogram using Euclidean distances via the UPGMA method. We also applied k-means clustering to determine the optimal number of clusters, using a K range from 1 to 10 with 1000 bootstrapping iterations, using the factoextra 1.0.7 and stats 4.3.0 packages in R. The significance of each morpho-phenological trait in the discrimination of populations/individuals was determined using eigenvalues from a PCA (Appendix A). The correlation between phenotypic data, geographical distribution, and genetic data was analyzed using the Mantel test with 1000 permutations from the vegan 2.6-4 package in R. Graphical representations were generated using the ggbiplot, ggplot2 3.4.3, and ggfortify 0.4.16 packages in R.

### 4.4. DNA Isolation and ISSR Protocol

Total genomic DNA was extracted from fresh young leaves of plants cultivated in pots, following the protocol described by Brugnoli et al. [16]. DNA quality was confirmed by assessing DNA integrity and verifying the absence of RNA contamination. Each DNA sample was standardized to 20 ng μL^−1^ for use in PCR amplification.

Sixteen ISSR primers were used for PCR amplification. The primers analyzed were the following: (AC)8T, (AC)8G, (AG)8T, (AG)8C, (AG)8GC, (AGAC)4GC, (ATG)5GA, (CA)8G, CAG(CA)7, (CT)8G, (CTC)6AC, (GA)8C, (GA)8T, (GA)8TC, (GAG)7AC, and (TC)8A. PCR reactions were conducted with a 25 µL final volume containing 20 ng of template DNA, 2.5 µL of 10× reaction buffer, 1.5 µL of 25 mM MgCl2, 1.5 µL of 5 µM primer, 1.25 µL of 2 mM dNTPs, 0.2 µL of Taq DNA polymerase (5 U µL^−1^), and H2O. DNA amplifications were performed in a T100 thermal cycler (Bio-Rad, Hercules, CA, USA) using the following thermal cycle: an initial denaturing step at 94 °C for 5 min; 40 cycles of 94 °C for 1 min, with primer annealing at a specific temperature for 45 s (Appendix A); 72 °C for 2 min; and a final extension at 72 °C for 5 min. PCR products were separated by electrophoresis in 2% agarose gels in 1X TAE buffer (1M Tris, 0.5M EDTA, acetic acid, and pH 8), at 60 V for 3,5 h, and stained with ethidium bromide (1 µg ml^−1^). The molecular profiles were visualized under UV light, photographed, and analyzed with the GelDoc-It Imaging System.

### 4.5. Molecular Diversity Within Populations

Molecular diversity within each population was calculated using the total number of bands (TBs), the number of locally common bands occurring in ≤50% populations (CBs), the percentage of polymorphic loci (PL%), and the number of private bands (PBs) per population, using GenAlEx 6 [104]. Genotypic variation was assessed using the number of effective genotypes (G), Nei’s diversity index (D), the evenness index (E), and the Shannon diversity index (H) under an independent mutation model with five mutational steps (M = 5) using GENOTYPE/GENODIVE see [105].

### 4.6. Differentiation Between Populations

The population structure for each species was estimated using AMOVA with significance testing based on 999 permutations, using the adegenet 2.1.10, poppr 2.9.4, and mmod 1.3.3 packages in R [106,107,108]. We calculated genetic divergence using the Rho_ST_ value, a multiallelic analog of FST for dominant markers and polyploids, and estimated gene flow using pairwise-FST with Nei`s distance among populations via the hierfstat 0.5-11 package in R [109].

Pairwise genetic distances among populations were estimated using Nei’s Distance with 999 permutations using the poppr 2.9.4 package in R. A clustering analysis was performed, and a UPGMA dendrogram was constructed using the package poppr 2.9.4 in R. The effective number of clusters was analyzed using k-means clustering with the adegenet 2.1.10 package in R (Appendix A). For the structure analysis, the LEA package was employed [110], which estimates admixture coefficients using sparse non-negative matrix factorization algorithms and provides STRUCTURE-like outputs. We used a loci-prior model with 20,000 iterations for a K range of 1 to 10 (K = [1,2,3,4,5,6,7,8,9,10]), with 50 replicates each (Appendix A). Identified clusters were then plotted into a map using the packages LEA 3.12.2 and mapplots 1.5.2 in R. Discriminant analysis of principal coordinates (DAPC) was conducted to visualize the distribution of genetic diversity between populations using the adegenet 2.1.10 and ade4 1.7-2.2 packages in R [111]. A Mantel test with 999 permutations was used to determine the geographical isolation among genetic clusters using the vegan 2.6-4 package, and significant isolated clusters were identified using the mpmcorrelogram 0.1-4 package in R [112] (Appendix A).

## 5. Conclusions

Our phenotypic analyses of four sexual *Paspalum* species, two outcrossing (*P. durifolium* and *P. ionanthum*) and two selfing (*P. regnellii* and *P. urvillei*), revealed low intrapopulation phenotypic variability and no significant differentiation among populations for both groups. In selfing species, homozygosity increases, leading to reduced intrapopulation variation and greater differentiation among populations. Conversely, allogamous species benefit from cross-pollination, maintaining phenotypic uniformity across populations. Despite expectations of greater differentiation among selfing species, overlapping phenotypic ranges were observed in both mating systems, suggesting that gene flow via pollen or seeds plays a significant role in these populations. In contrast, the apomictic species *P. intermedium* showed low intrapopulation variation but high differentiation between populations, indicating a fixation of well-adapted genotypes through genetic drift and local adaptation, leading to distinct morphological clustering.

Outcrossing species like *P. durifolium* and *P. ionanthum* exhibited high molecular and genotypic variation within populations, while selfing species such as *P. regnellii* and *P. urvillei* displayed comparatively lower genetic diversity. Differences in polymorphism and genotypic diversity among selfing species could be attributed to varying rates of allogamy and geographic distribution. Selfing species generally exhibit greater population differentiation due to limited pollen migration, increasing genetic drift and the fixation of alleles in isolated populations. Indeed, higher values of genetic differentiation were observed in selfing species compared to outcrossing ones. However, *P. ionanthum*, despite being strictly outcrossing, exhibited notable genetic differentiation among populations. This highlights the role of geographic range in genetic diversity, as isolation by distance was evident in both selfing and outcrossing species, indicating the importance of pollen and seed dispersal in shaping population structure.

In apomictic populations of *P. intermedium*, low molecular and genotypic variation led to mostly uniclonal populations with minimal polymorphism compared to the selfing and outcrossing species. This lack of gene flow and limited sexual reproduction resulted in a highly structured genetic landscape, with significant differentiation among populations. The presence of distinct dominant genotypes across populations suggested local adaptation to specific environmental conditions. Apomictic species like *P. intermedium* experience high population differentiation and limited genetic exchange, leading to a strong spatial genetic structure. These findings underscore the complexities of genetic diversity in apomictic species, emphasizing the influence of local adaptation and restricted gene flow.

While these findings provide valuable insights, future studies with larger sample sizes, broader geographical scopes, and additional molecular markers are necessary to validate these patterns and better understand the interplay between reproductive modes and genetic diversity in polyploid species of *Paspalum*. Furthermore, understanding the interplay between reproductive modes and genetic diversity can reduce time and money costs in conservation and breeding strategies for *Paspalum* spp.

## Figures and Tables

**Figure 1 plants-14-00476-f001:**
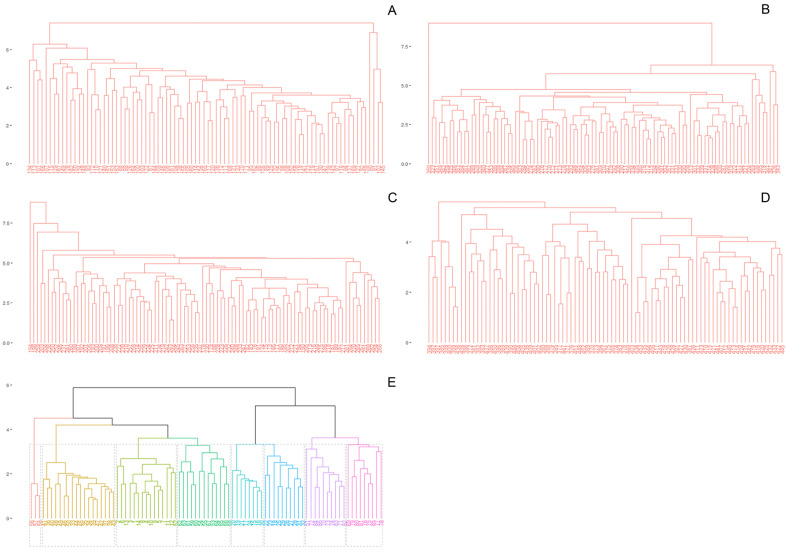
UPGMA trees reflecting optimal number of clusters using phenotypic traits. (**A**) *P. durifolium.* (**B**) *P. ionanthum.* (**C**) *P. regnellii.* (**D**) *P. urvillei.* (**E**) *P. intermedium*. Clusters within each tree are according to the k-means algorithm and indicated by different colors.

**Figure 2 plants-14-00476-f002:**
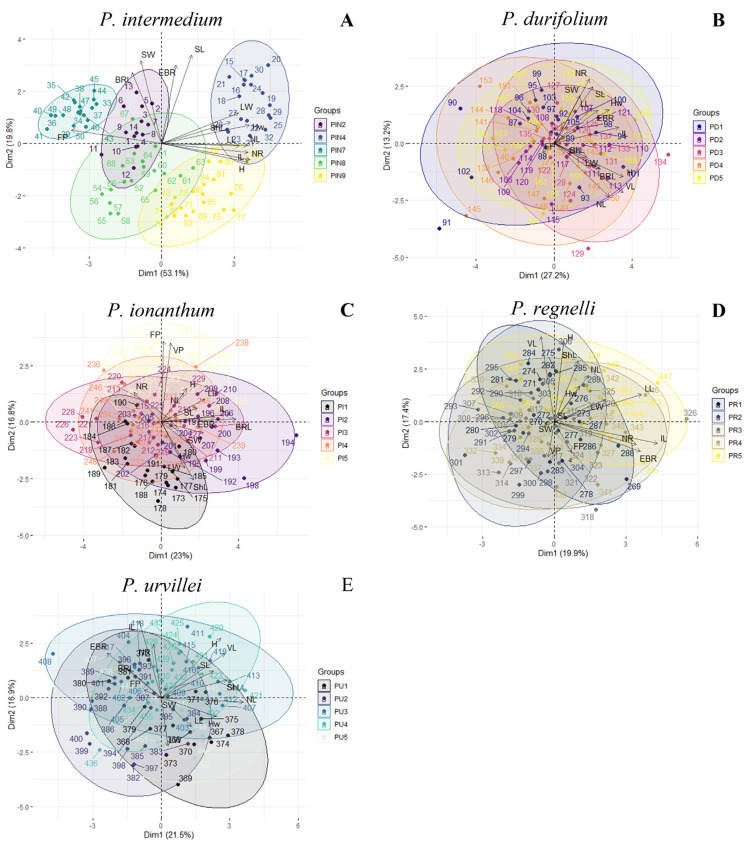
PCA analysis for each species using morpho-phenological traits. (**A**) *P. intermedium.* (**B**) *P. durifolium.* (**C**) *P. ionanthum.* (**D**) *P. regnellii.* (**E**) *P. urvillei*.

**Figure 3 plants-14-00476-f003:**
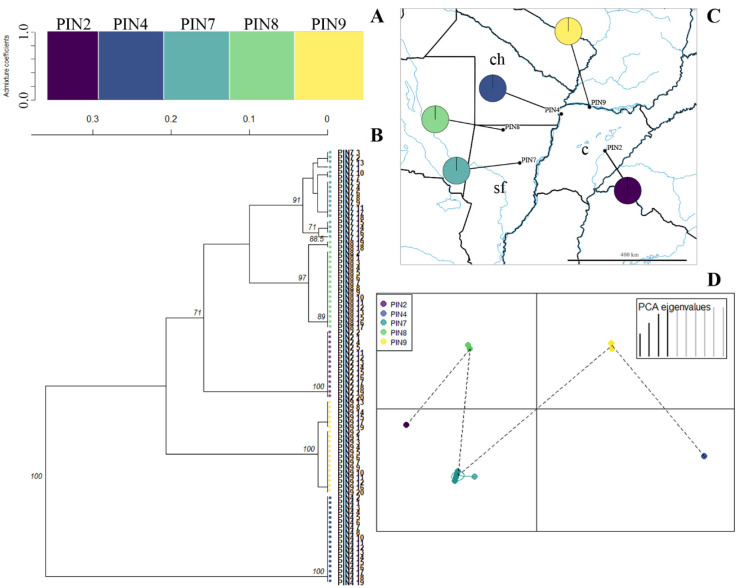
Genetic clusters inferred from ISSR in *Paspalum intermedium* populations. (**A**). Bayesian clustering of all individuals (K = 5). Bars represent individuals with the proportion of admixture in different colors. Light grey lines separate populations. (**B**). Unrooted UPGMA dendrogram constructed using Nei’s distance with 999 permutations. Only bootstrap values above 70 are shown. Color codes correspond with the DAPC legend (see D). (**C**). Geographical distribution of *P. intermedium* populations. Color codes in pie charts indicate the genetic composition (admixture coefficients) regarding the Bayesian clustering analysis of each population inferred from ISSRs (see A). (**D**). Discriminant analysis of principal coordinates of genetic variation. Lines represent the genetic distance of each individual from the centroid of each ellipse, and ellipses represent 95% dispersion of individual genetic variation from each population. Dashed lines connect populations that are more similar. c: Corrientes, ch: Chaco, sf: Santa Fe.

**Figure 4 plants-14-00476-f004:**
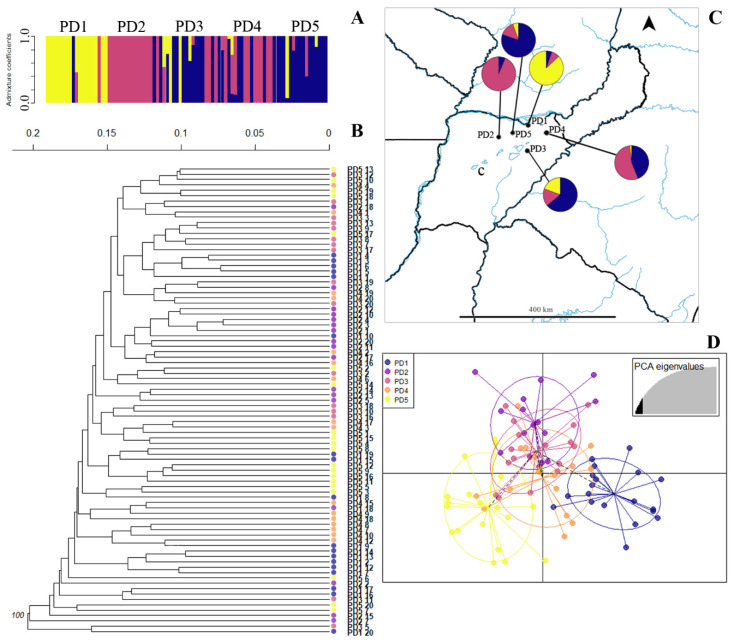
Genetic clusters inferred from ISSR in *Paspalum durifolium* populations. (**A**). Bayesian clustering of all individuals (K = 3). Bars represent individuals with the proportion of admixture in different colors. Light grey lines separate populations. (**B**). Unrooted UPGMA dendrogram constructed using Nei’s distance with 999 permutations. Only bootstrap values above 70 are shown. Color codes correspond with the DAPC legend (see D). (**C**). Geographical distribution of *P. durifolium* populations. Color codes in pie charts indicate the genetic composition (admixture coefficients) regarding the Bayesian clustering analysis of each population inferred from ISSRs (see A). (**D**). Discriminant analysis of principal coordinates of genetic variation. Lines represent the genetic distance of each individual from the centroid of each ellipse, and ellipses represent 95% dispersion of individual genetic variation from each population. Dashed lines connect populations that are more similar. c: Corrientes.

**Figure 5 plants-14-00476-f005:**
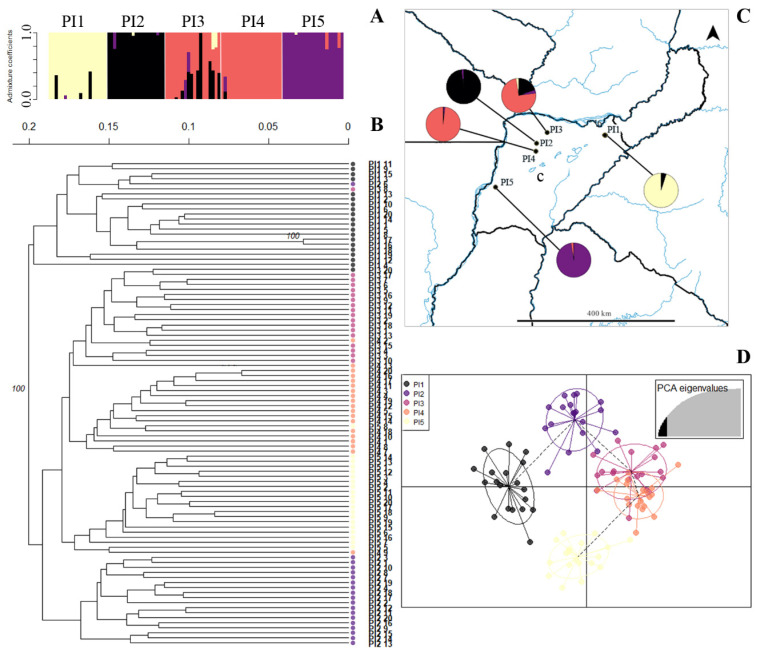
Genetic clusters inferred from ISSR in *Paspalum ionanthum* populations. (**A**). Bayesian clustering of all individuals (K = 4). Bars represent individuals with the proportion of admixture in different colors. Light grey lines separate populations. (**B**). Unrooted UPGMA dendrogram constructed using Nei’s distance with 999 permutations. Only bootstrap values above 70 are shown. Color codes correspond with the DAPC legend (see D). (**C**). Geographical distribution of *P. ionanthum* populations. Color codes in pie charts indicate the genetic composition (admixture coefficients) regarding the Bayesian clustering analysis of each population inferred from ISSRs (see A). (**D**). Discriminant analysis of principal coordinates of genetic variation. Lines represent the genetic distance of each individual from the centroid of each ellipse, and ellipses represent 95% dispersion of individual genetic variation from each population. Dashed lines connect populations that are more similar. c: Corrientes.

**Figure 6 plants-14-00476-f006:**
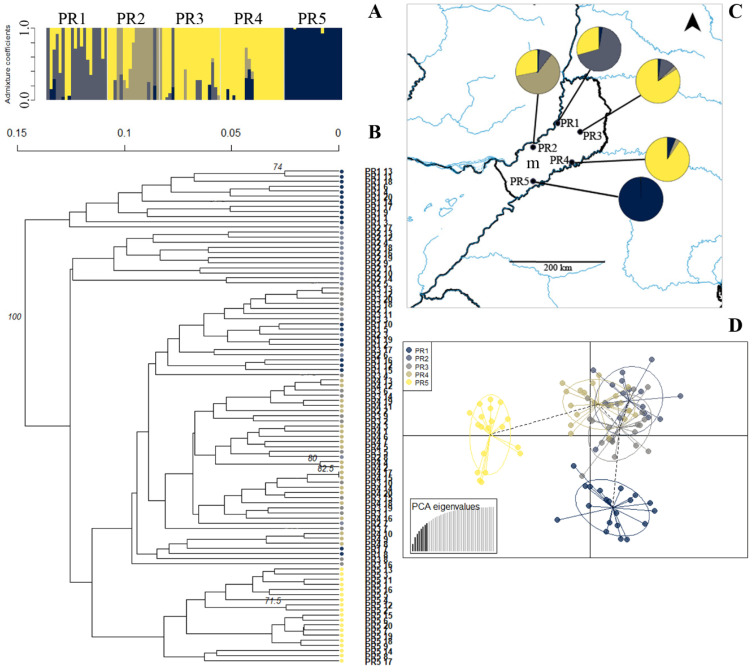
Genetic clusters inferred from ISSR in *Paspalum regnellii* populations. (**A**). Bayesian clustering of all individuals (K = 4). Bars represent individuals with the proportion of admixture in different colors. Light grey lines separate populations. (**B**). Unrooted UPGMA dendrogram constructed using Nei’s distance with 999 permutations. Only bootstrap values above 70 are shown. Color codes correspond with the DAPC legend (see D). (**C**). Geographical distribution of *P. regnellii* populations. Color codes in pie charts indicate the genetic composition (admixture coefficients) regarding the Bayesian clustering analysis of each population inferred from ISSRs (see A). (**D**). Discriminant analysis of principal coordinates of genetic variation. Lines represent the genetic distance of each individual from the centroid of each ellipse, and ellipses represent 95% dispersion of individual genetic variation from each population. Dashed lines connect populations that are more similar. m: Misiones.

**Figure 7 plants-14-00476-f007:**
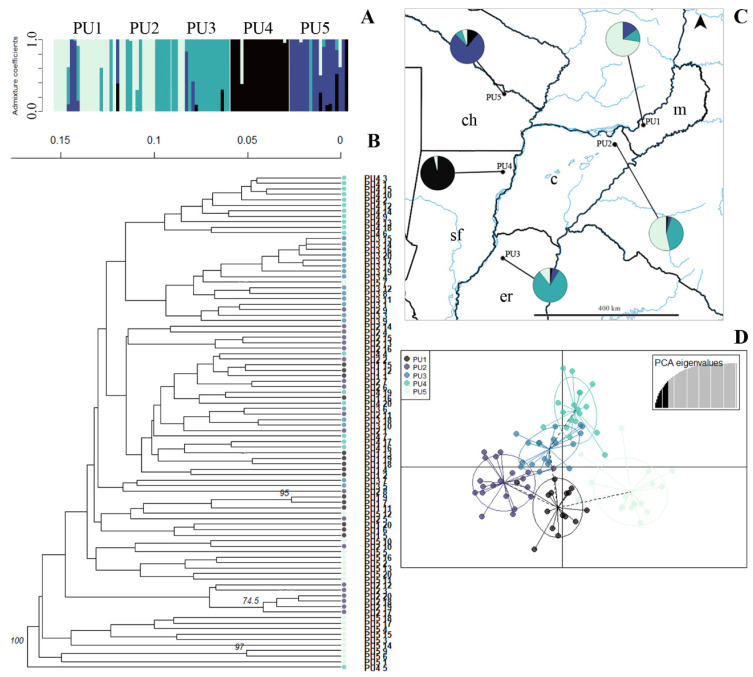
Genetic clusters inferred from ISSR in *Paspalum urvillei* populations. (**A**). Bayesian clustering of all individuals (K = 4). Bars represent individuals with the proportion of admixture in different colors. Light grey lines separate populations. (**B**). Unrooted UPGMA dendrogram constructed using Nei’s distance with 999 permutations. Only bootstrap values above 70 are shown. Color codes correspond with the DAPC legend (see D). (**C**). Geographical distribution of *P. urvillei* populations. Color codes in pie charts indicate the genetic composition (admixture coefficients) regarding the Bayesian clustering analysis of each population inferred from ISSRs (see A). (**D**). Discriminant analysis of principal coordinates of genetic variation. Lines represent the genetic distance of each individual from the centroid of each ellipse, and ellipses represent 95% dispersion of individual genetic variation from each population. Dashed lines connect populations that are more similar. c: Corrientes, ch: Chaco, er: Entre Rios, m: Misiones, sf: Santa Fe.

**Table 1 plants-14-00476-t001:** Mean and standard deviation for each phenotypic trait in five *Paspalum* species.

Species	Pop	n	H	Hw	LL	LW	ShL	NL	IL	BRL	EBR	NR	VL	SL	SW	VP	FP
*P. intermedium*	PIN2	14	187.9 ± 17.2	100.7 ± 9.2	34.2 ± 4.2	1.2 ± 0.1	26.4 ± 3.0	26.3 ± 3.2	25 ± 1.5	9.5 ± 1.0	110.6 ± 9.8	39.5 ± 2.5	16.5 ± 10.7	2.7 ± 0.2	1.6 ± 0.0	81.2 ± 9.1	114.8 ± 5.6
	PIN4	18	228.1 ± 17.2	131.1 ± 9.6	45.8 ± 5.5	1.9 ± 0.1	40.2 ± 5.6	36.6 ± 3.7	37.3 ± 2.7	9.0 ± 0.9	145.9 ± 36.4	77.6 ± 9.8	205.6 ± 11.8	3.0 ± 0.1	1.5 ± 0.0	62.0 ± 6.7	77.4 ± 3.5
	PIN7	18	110.3 ± 21.1	71.4 ± 14.1	23.6 ± 2.9	1.2 ± 0.1	29.4 ± 2.4	11.0 ± 4.1	22.4 ± 2.4	10.9 ± 1.1	135.7 ± 10.9	33.4 ± 5.1	103.3 ± 18.9	2.7 ± 0.1	1.5 ± 0.0	112 ± 12.5	148.2 ± 8.7
	PIN8	18	173.8 ± 31.3	85.6 ± 9.9	35.8 ± 7.5	1.4 ± 0.3	29.4 ± 5.5	23.9 ± 3.6	26.9 ± 3.5	8.2 ± 1.6	105.7 ± 21.5	45.9 ± 4.8	163.9 ± 29.3	2.5 ± 0.1	1.3 ± 0.1	92.1 ± 12.6	126.8 ± 16.2
	PIN9	18	225.7 ± 25.6	106.8 ± 14.7	38.6 ± 10.2	1.4 ± 0.3	34.1 ± 3.8	28.3 ± 5.2	35.4 ± 4.8	8.1 ± 1.1	108.5 ± 14.8	71.2 ± 4.7	210.3 ± 19.1	2.4 ± 0.1	1.2 ± 0.1	66.1 ± 6.0	81.7 ± 4.2
*P. durifolium*	PD3	16	118.1 ± 17.4	57.9 ± 6.9	24.4 ± 6.4	0.9 ± 0.1	11.7 ± 2.9	23.2 ± 5.3	19.7 ± 2.4	8.6 ± 1.0	90.2 ± 11.9	19.4 ± 3.3	100 ± 14.6	3.0 ± 0.1	1.5 ± 0.1	62.4 ± 6.2	48.5 ± 6.5
	PD4	16	111.3 ± 21.3	53.6 ± 5.7	20.1 ± 3.0	0.8 ± 0.1	10.5 ± 2.3	18.9 ± 6.1	18.7 ± 2.3	7.7 ± 1.2	84.2 ± 14.6	17.2 ± 3.6	88.6 ± 19.1	3.0 ± 0.1	1.4 ± 0.1	57.4 ± 8.1	74.9 ± 6.7
	PD5	19	114 ± 16.8	58.3 ± 6.3	25.6 ± 5.2	0.8 ± 0.1	10.7 ± 2.4	19.5 ± 4.6	18.7 ± 2.3	7.8 ± 1.1	104.5 ± 18.5	19.5 ± 3.0	89.9 ± 13.5	2.9 ± 0.1	1.4 ± 0.1	60.1 ± 4.2	74.4 ± 4.8
	PD1	19	107.5 ± 19.6	58.7 ± 9.7	25.3 ± 4.8	0.8 ± 0.1	11.6 ± 2.9	16.5 ± 5.7	18.7 ± 2.0	7.4 ± 1.3	93.3 ± 18.9	17.7 ± 4.0	87.9 ± 16.6	2.9 ± 0.1	1.4 ± 0.1	68.3 ± 6.4	81.9 ± 5.6
	PD2	16	118.7 ± 14.2	60.2 ± 12.0	25.7 ± 4.9	0.8 ± 0.1	9.8 ± 1.4	21.8 ± 4.4	18.9 ± 2.4	8.2 ± 1.4	98.6 ± 15.6	18.5 ± 3.1	98.6 ± 11.9	2.9 ± 0.1	1.4 ± 0.1	63.6 ± 5.2	76.9 ± 4.3
*P. ionanthum*	PI3	19	109.7 ± 21.2	67.7 ± 7.4	29.3 ± 5.0	0.7 ± 0.1	12.1 ± 2.0	25.8 ± 5.3	10.4 ± 1.7	9.3 ± 1.4	69.9 ± 12.3	2.1 ± 0.2	94.5 ± 21.1	4.2 ± 0.3	2 ± 0.0	62.1 ± 8.5	82.4 ± 6.2
	PI4	20	121.0 ± 18.8	69.1 ± 7.1	31.4 ± 5.2	0.8 ± 0.1	8.4 ± 1.5	25.8 ± 4.7	11.1 ± 1.3	9.5 ± 1.4	70.5 ± 14.4	2.3 ± 0.4	108.0 ± 16.8	4.4 ± 0.3	2.0 ± 0.0	56.6 ± 9.4	75.8 ± 7.4
	PI5	19	126.3 ± 21.0	66.3 ± 9.7	31.0 ± 3.6	0.8 ± 0.1	9.0 ± 2.1	28.5 ± 5.3	11.8 ± 1.6	10.1 ± 1.4	73.3 ± 11.4	2.2 ± 0.3	116.1 ± 20.9	4.4 ± 0.3	2.0 ± 0.1	77.3 ± 8.8	92.8 ± 8.9
	PI1	19	96.5 ± 6.7	67.8 ± 7.7	26.1 ± 3.0	0.8 ± 0.1	13.9 ± 2.4	23.0 ± 2.3	10.7 ± 1.3	9.7 ± 1.1	72.3 ± 11.2	2.0 ± 0.1	88.7 ± 3.4	4.2 ± 0.2	2.0 ± 0.0	51.6 ± 10.8	74.0 ± 9.3
	PI2	19	102.1 ± 20.8	72.3 ± 7.6	32.3 ± 5.7	0.8 ± 0.1	15.3 ± 3.1	22.0 ± 6.4	11.9 ± 1.8	11.0 ± 1.8	82.1 ± 11.9	2.0 ± 0.1	89.3 ± 17.6	4.6 ± 0.2	2.2 ± 0.2	68.3 ± 8.2	83.3 ± 7.6
*P. regnellii*	PR2	18	183.3 ± 12.8	92.8 ± 7.6	34.5 ± 5.3	2.3 ± 0.2	14.9 ± 3.2	18.8 ± 2.6	12.4 ± 1.8	10.3 ± 1.2	144.9 ± 23.3	9.4 ± 1.3	164.1 ± 16.2	2.5 ± 0.0	1.5 ± 0.0	133.9 ± 16.0	165.0 ± 2.5
	PR1	20	185.6 ± 11.7	95.5 ± 2.8	33.8 ± 3.5	2.3 ± 0.2	18.0 ± 2.6	17.9 ± 2.00	14.1 ± 2.2	10.7 ± 1.1	179.8 ± 29.0	10.6 ± 2.2	163.9 ± 16.8	2.5 ± 0.0	1.5 ± 0.0	116.2 ± 8.8	161.2 ± 1.8
	PR3	20	185 ± 10.9	99.0 ± 9.8	34.7 ± 5.4	2.3 ± 0.3	15.8 ± 1.9	19.2 ± 2.0	13.8 ± 2.3	11.0 ± 1.7	164.0 ± 33.2	10.3 ± 2.1	154.9 ± 14.6	2.5 ± 0.0	1.5 ± 0.0	142.3 ± 17.7	169.5 ± 5.7
	PR4	20	184.5 ± 12.2	101.2 ± 7.2	35 ± 5.3	2.4 ± 0.3	14.3 ± 2.2	19.0 ± 3.0	14.4 ± 2.3	11.7 ± 1.6	163.4 ± 25.0	11.0 ± 2.1	157.4 ± 12.9	2.5 ± 0.0	1.5 ± 0.0	144.2 ± 12.7	167.6 ± 4.4
	PR5	19	197.9 ± 8.5	112.1 ± 24.2	37.9 ± 5.5	2.2 ± 0.1	16.0 ± 1.7	20.3 ± 2.4	15.5 ± 2.4	12.2 ± 1.3	167.9 ± 24.2	11.8 ± 2.9	172.6 ± 11.9	2.5 ± 0.0	1.5 ± 0.0	131.4 ± 16.0	165.4 ± 4.9
*P. urvillei*	PU1	16	199.1 ± 8.4	135.3 ± 10.4	43.3 ± 5.1	1.6 ± 0.1	20.8 ± 2.1	20.4 ± 2.1	28.4 ± 4.9	11.4 ± 2.0	139.0 ± 24.0	14.4 ± 3.6	177.6 ± 18.3	2.9 ± 0.1	1.5 ± 0.0	43 ± 10.1	70.1 ± 3.6
	PU3	18	216.4 ± 11.2	118.1 ± 9.6	36 ± 2.7	1.2 ± 0.1	20.2 ± 2.4	19.4 ± 3.0	29.4 ± 3.3	10.5 ± 1.8	145.5 ± 23.1	14.8 ± 2.8	191.0 ± 13.6	2.9 ± 0.1	1.5 ± 0.0	52.8 ± 16	82.2 ± 14.8
	PU4	17	213.1 ± 15.6	117 ± 9.6	34.4 ± 5.4	1.2 ± 0.1	22.7 ± 2.5	20.7 ± 2.1	31.4 ± 3.2	10.4 ± 1.1	133.7 ± 13.8	15.8 ± 1.6	183.0 ± 16.3	2.8 ± 0.1	1.5 ± 0.0	56.6 ± 23.2	85.6 ± 18
	PU5	17	206.7 ± 12.2	124.9 ± 11.4	39.7 ± 5.6	1.4 ± 0.2	20.4 ± 2.4	19.5 ± 2.8	29.5 ± 4.0	11.1 ± 2.1	139.7 ± 20.6	14.1 ± 1.9	177.2 ± 17.7	2.8 ± 0.1	1.5 ± 0.0	72.1 ± 25	97.4 ± 26.4
	PU2	20	196 ± 13.2	119 ± 12.5	35.8 ± 4.4	1.4 ± 0.2	19.6 ± 2.0	18.3 ± 2.5	28.5 ± 5.8	11.8 ± 2.2	144.6 ± 16.9	12.9 ± 2.4	171.2 ± 15.7	2.7 ± 0.2	1.5 ± 0.0	47.8 ± 17.3	79.6 ± 17.5

Abbreviations, n: number of individuals, plant height with (H; cm) and without flowering culms (Hw; cm), 2nd leaf blade length (LL; cm) and width (LW; cm), 2nd leaf sheath length (ShL; cm), 2nd internode length (NL; cm), inflorescence length (IL; cm). n: number of individuals, basal raceme length (BRL; cm), number of spikelets in the basal raceme (EBR), raceme number (NR), flowering culms length with inflorescences (VL; cm), spikelet length (SL; mm) and width (SW; mm), extension of the vegetative period (VP; days) and the flowering period (FP; days).

**Table 2 plants-14-00476-t002:** Molecular and genotypic diversity indexes in tetraploid populations of five *Paspalum* species.

Species	Pop	*n*	Molecular Diversity Indexes				Genotypic Diversity Indexes			
			*TB*	*CB*	*%PL*	*PB*	*G*	*D*	*E*	*H*
*P. intermedium*	PIN2	14	81	3	0	1	1	0	1	0
	PIN4	18	75	3	0	3	1	0	1	0
	PIN7	18	78	1	4.4	0	1	0	1	0.022
	PIN8	18	78	1	2.2	0	1	0	1	0.008
	PIN9	19	77	2	1.1	0	1	0	1	0.007
*P. durifolium*	PD1	19	123	0	77.4	0	19	1	1	1.28
	PD2	16	123	0	80.7	0	16	1	1	1.20
	PD3	16	123	0	79.8	0	16	1	1	1.20
	PD4	16	123	0	75.8	0	16	1	1	1.20
	PD5	20	123	0	79.8	0	20	1	1	1.30
*P. ionanthum*	PI1	19	131	0	76.3	0	19	1	1	1.27
	PI2	19	132	2	76.3	0	19	1	1	1.27
	PI3	19	134	1	81.3	0	19	1	1	1.27
	PI4	19	134	1	80.6	1	19	1	1	1.27
	PI5	20	131	0	74.8	0	20	1	1	1.30
*P. regnellii*	PR1	20	120	1	26.2	0	19	0.99	0.96	1.27
	PR2	18	120	0	25.4	0	18	1	1	1.26
	PR3	20	120	0	21.3	0	13	0.91	0.59	1.01
	PR4	20	116	0	14.8	0	8	0.77	0.47	0.72
	PR5	19	115	1	15.6	0	9	0.81	0.48	0.79
*P. urvillei*	PU1	15	100	0	49.0	0	15	1	1	1.18
	PU2	20	101	0	53.9	0	20	1	1	1.30
	PU3	19	100	0	42.2	0	19	1	1	1.28
	PU4	18	100	0	46.1	0	18	1	1	1.26
	PU5	18	102	0	55.9	0	18	1	1	1.26

*n*: number of individuals; *TB*: total number of bands; *CB*: no. of common bands (≤50%); *%PL*: percentage of polymorphic loci, *PB*: private band number; *G*: effective number of genotypes; *D*: Nei`s genetic diversity index; *E*: evenness index; *H*: Shannon–Wiener index.

**Table 3 plants-14-00476-t003:** AMOVA test for *Paspalum* species based on ISSR data, respectively.

Species	Source	df	SS	MS	Est. Var	Rho_ST_	*p*
*P. intermedium*	Among Pops	4	534.5	133.6	7.7	0.851	0.001
	Within Pops	82	20.2	0.25	0.3		
*P. durifolium*	Among Pops	4	188.6	47.2	1.7	0.084	0.001
	Within Pops	82	1490.7	18.1	18.2		
*P. ionanthum*	Among Pops	4	380.7	95.2	4.0	0.18	0.001
	Within Pops	91	1666.8	18.3	18.3		
*P. regnellii*	Among Pops	4	160.5	40.1	1.8	0.29	0.001
	Within Pops	92	415.8	4.5	4.5		
*P. urvillei*	Among Pops	4	200.4	50.1	2.3	0.20	0.001
	Within Pops	85	793.8	9.3	9.3		

Abbreviations, df: degree of freedom, SS: square sums, MS: mean of the squares, Est. Var: estimation of variation, Rho_ST_: analog of F_ST_ for polyploids, and *p*: *p*-value.

**Table 4 plants-14-00476-t004:** Pairwise fixation index (AMOVA-derived FST) between populations within each species of *Paspalum* calculated with 999 permutations on ISSR data.

	** *P. intermedium* **										
**Pop**	PIN2	PIN4	PIN7	PIN8	PIN9						
PIN2	-	***	***	***	***						
PIN4	**0.12**	-	***	***	***						
PIN7	0.07	**0.13**	-	***	***						
PIN8	0.07	**0.13**	0.04	-	***						
PIN9	0.09	0.09	0.08	0.08	-						
	** *P. durifolium* **						** *P. ionanthum* **				
**Pop**	PD1	PD2	PD3	PD4	PD5	**Pop**	PI1	PI2	PI3	PI4	PI5
PD1	-	**	**	**	***	PI1	-	n.s.	n.s.	n.s.	***
PD2	0.02	-	n.s.	n.s.	***	PI2	0.04	-	n.s.	n.s.	***
PD3	0.02	0.02	-	n.s.	***	PI3	0.04	0.03	-	n.s.	***
PD4	0.02	0.01	0.01	-	***	PI4	0.05	0.04	0.03	-	***
PD5	0.02	0.02	0.01	0.02	-	PI5	0.05	0.05	0.04	0.04	-
	** *P. regnellii* **						** *P. urvillei* **				
**Pop**	PR1	PR2	PR3	PR4	PR5	**Pop**	PU1	PU2	PU3	PU4	PU5
PR1	-	n.s.	***	***	***	PU1	-	n.s.	n.s.	n.s.	n.s.
PR2	0.02	-	n.s.	***	***	PU2	0.02	-	n.s.	n.s.	n.s.
PR3	0.01	0.01	-	***	***	PU3	0.02	0.03	-	n.s.	n.s.
PR4	0.02	0.01	0.01	-	***	PU4	0.03	0.03	0.03	-	n.s.
PR5	0.03	0.02	0.02	0.02	-	PU5	0.02	0.03	0.03	0.03	-

*p*-value: ** 0.01, *** 0.001, n.s. not significant.

**Table 5 plants-14-00476-t005:** *Paspalum* species, population identification (pop), number of plants (N), reproductive modes and mating systems (rm), coordinates of each collection site and habitat type.

Species	Pop	N	rm ⁋	Location and Habitat Type
*P. durifolium*	PD1	19	S * ss	C, 27.627033° S, 56.747883° W. U.
	PD2	18	S ss	C, 27.931683° S, 57.5166° W. U.
	PD3	17	S * ss	C, 28.290583° S, 56.770316° W. D.
	PD4	18	S * ss	C, 27.825216° S, 56.2549° W. U.
	PD5	20	S * ss	C, 27.560766° S, 57.153066° W. U.
*P. ionanthum*	PI1	20	S * ss	C, 27.821083° S, 56.27213° W. D.
	PI2	20	S * ss	C, 28.237416° S, 58.0542° W. U.
	PI3	16	S ss	C, 27.753416° S, 57.76096° W. U.
	PI4	20	S ss	C, 28.033733° S, 58.0362° W. U.
	PI5	20	S ss	C, 29.1651° S, 59.096583° W. D.
*P. regnellii*	PR1	20	S sf	M, 26.54153° S, 54.7251° W. D.
	PR2	19	S sf	M, 27.052833° S, 55.25066° W. U.
	PR3	20	S sf	M, 26.7271° S, 54.247416° W. D.
	PR4	20	S sf	M, 27.374017° S, 54.42505° W. U.
	PR5	19	S sf	M, 27.77713° S, 55.24996° W. U.
*P. urvillei*	PU1	17	S sf	M, 27.276883° S, 55.46225° W. D.
	PU2	20	S sf	C, 27.821083° S, 56.272133° W. D.
	PU3	20	S sf	ER, 31.02038° S, 59.420366° W. D.
	PU4	20	S sf	SF, 28.59555° S, 59.41675° W. D.
	PU5	18	S sf	Ch, 26.40195° S, 59.37876° W. D.
*P. intermedium*	PIN2	14	A ap ps sf	C, 28.87166° S, 57.26405° W. D.
	PIN4	18	A ap ps sf	C, 27.615016° S, 58.75398° W. U.
	PIN7	18	A ap ps sf	SF, 29.29074° S, 60.17697° W. U.
	PIN8	18	A ap ps sf	SF, 28.15866° S, 60.74791° W. D.
	PIN9	19	A ap ps sf	C, 27.38394° S, 57.78926° W. U.

⁋ data from Schedler et al. [48] except for *P. intermedium*. S: sexuality; ss: self-sterile; sf: self-fertile; A: apomixis; ap: apospory; ps: pseudogamy. * A low proportion of aposporic embryo sacs were observed in these populations, but no apomictic seeds were observed [48]. Argentina’s provinces: C: Corrientes, Ch: Chaco, ER: Entre Rios, M: Misiones, and SF: Santa Fe. Habitat types: D: disturbed habitats and U: undisturbed habitats.

## Data Availability

All the data presented in this study are available in this article and in the Appendix A, and binary (ISSR) and phenotypic arrays are uploaded in http://hdl.handle.net/11336/253318.

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
