# Peer review of "The Role of Reproductive Modes in Shaping Genetic Diversity in Polyploids: A Comparative Study of Selfing, Outcrossing, and Apomictic Paspalum Species"

_plants, 2025, doi:10.3390/plants14030476_

Round 1
Reviewer 1 Report
Comments and Suggestions for Authors
This is a nice paper. It presents good work that looks at Paspalum spp diversity and relates this well to the reproductive system of the species studied. It will provide valuable information to efforts to breed high-performing Paspalum grass cultivars.
I suggest a few things need to be fixed:
line 42; no '
from line 55; the text says there are two types of apomixis, but then only describes the genetic consequences of one of the types
line 91; suggest replace "grass" with Paspalum
line 96; (journal not author problem) hyphenate "Although" as "Al-though", not "Alt-hough"
line 122 (and following examples); "References" should be "Abbreviations"
line 125 "Continued" not "Continue"
Figure 1; this is much too low resolution, there is no way anyone can read the identifications along the base of each tree
line 158 and following: "Mantel's" has the apostrophe in a reverse slope to normal (probably a journal, not author problem)
Figure 3 and following figures; these are also a bit low resolution for reading
Author Response
We sincerely thank you for your valuable comments and constructive suggestions. We deeply appreciate the time and effort you dedicated to evaluating our manuscript. Your feedback has significantly contributed to improving the clarity, depth, and overall quality of our work. We have carefully addressed all your comments and, where applicable, provided a detailed rationale for our decisions. Your insightful critiques have greatly enhanced the scientific rigour of our manuscript, and we are truly grateful for the opportunity to incorporate these meaningful improvements.
Below we summarize responses to each specific point raised:
This is a nice paper. It presents good work that looks at Paspalum spp. diversity and relates this well to the reproductive system of the species studied. It will provide valuable information to efforts to breed high-performing Paspalum grass cultivars.
Reviewer: I suggest a few things need to be fixed:
- line 42; no '
Response: We removed the apostrophe.
- from line 55; the text says there are two types of apomixis, but then only describes the genetic consequences of one of the types
Response: Thanks for this observation, we probably deleted the description of sporophytic apomixis in the authors edition phase. Thus, we added the description of sporophytic apomixis. Genetic consequences of apomixis are the same regardless of the type (i.e. clonal progeny, low genotypic diversity, high heterozygosity, high mutation load, etc).
The following sentence was added:
“In sporophytic apomixis, also called polyembryony, an embryo is originated from the nucellus or the integuments independently of the of the gametophytic development [3].”
- line 91; suggest replace "grass" with Paspalum
Response: We replaced grass with Paspalum.
- line 96; (journal not author problem) hyphenate "Although" as "Al-though", not "Alt-hough"
Response: We added a high hyphen to fix this.
- line 122 (and following examples); "References" should be "Abbreviations"
Response: We replaced references with abbreviations.
- line 125 "Continued" not "Continue"
Response: We added the missing d.
- Figure 1; this is much too low resolution, there is no way anyone can read the identifications along the base of each tree
Response: Thanks for the observation. However, we uploaded the 300 dpi images as separate files to see details regarding plants ID, those inserted in the text had low resolution (100 dpi) due to the size restriction in the upload Journal page.
Nevertheless, we modified the display of Figure 1 in order to size up the plants ID.
- line 158 and following: "Mantel's" has the apostrophe in a reverse slope to normal (probably a journal, not author problem)
Response: We removed the apostrophe and s in those Mantel that had it.
- Figure 3 and following figures; these are also a bit low resolution for reading
Response: As mentioned above, the final manuscript will have 300 dpi images, nevertheless we will try to improve the reading of plants and pop ID.

Reviewer 2 Report
Comments and Suggestions for Authors
In general, the authors have done a good job presenting the objectives, results and discussion of their research study.
I have the following comments for the improvement of this manuscript.
First, the title is very general and does not convey any scientific meaning. Please revise your title according to the results of your study.
Please discuss the innovations of your study. What are the new pieces of information that the study revealed and that other scientists would benefit from?
My main question is the following. In the Materials and Methods, the authors report that they have used five populations, with approximately 20 plants per population. In the Tables, there are from 14 to 20 individual plants per population. I believe that this number of plants (14-20) may not be adequate to accurately study and discuss phenotypic variability and differentiation among and within populations.
Why did the authors choose only 14-20 plants? I think that a minimum of 30 plants per population is required in order to make precise conclusions on population variables.
There are some grammatical errors throughout, so please make sure you double check your manuscript for these.
Author Response
We thank you for your insightful comments and constructive suggestions. We greatly appreciate the time and effort you have invested in evaluating our manuscript. Your feedback has improved key areas of the manuscript, enhancing the clarity and depth of our work. We have addressed your comments wherever possible and provided a rationale where necessary. Your thoughtful critiques have greatly contributed to the quality and rigor of our manuscript, and we are grateful for the opportunity to incorporate these improvements.
Below we summarize responses to each specific point raised:
Reviewer: In general, the authors have done a good job presenting the objectives, results and discussion of their research study.
I have the following comments for the improvement of this manuscript.
- First, the title is very general and does not convey any scientific meaning. Please revise your title according to the results of your study.
Response: We suggest the following new title: The role of reproductive modes in shaping genetic diversity in polyploids: a comparative study of selfing, outcrossing, and apomictic Paspalum species.
- Please discuss the innovations of your study. What are the new pieces of information that the study revealed and that other scientists would benefit from?
Response: Unlike previous studies, we demonstrated that both selfing (P. regnellii and P. urvillei) and outcrossing species (P. durifolium and P. ionanthum) exhibit similar patterns of phenotypic diversity, challenging the conventional assumptions on the influence of mating systems in phenotypic variation in these species. Our findings revealed that population structure in these Paspalum species is a reflection on gene flow (either by pollen or seeds) among populations.
Also, we provided evidence on how apomixis (P. intermedium) leads to uniclonal populations with high interpopulation differentiation driven by genetic drift and local adaptation. This study highlights the importance of doing prior genetic analysis of multiple populations in order to design adjusted germplasm conservation strategies and genotype selection for future breeding programs.
Furthermore, understanding the interplay between reproductive modes and genetic diversity can reduce time and money costs in conservation and breeding strategies for Paspalum spp.
We include the following statement in the Abstract:
“This is the first report about genetic diversity in populations of sexual allopolyploid species of Paspalum.”
And we include the following statements in the Discussion:
“Morpho-phenological analyses like these highlight the importance of doing prior assessments of multiple populations. These studies help designing germplasm conservation programs that preserve all phenotypic clusters, besides helping in the selection of contrasting phenotypes for future breeding programs.”
And this sentence in the conclusion section:
“Furthermore, understanding the interplay between reproductive modes and genetic diversity can reduce time and money costs in conservation and breeding strategies for Paspalum spp.”
- My main question is the following. In the Materials and Methods, the authors report that they have used five populations, with approximately 20 plants per population. In the Tables, there are from 14 to 20 individual plants per population. I believe that this number of plants (14-20) may not be adequate to accurately study and discuss phenotypic variability and differentiation among and within populations. Why did the authors choose only 14-20 plants? I think that a minimum of 30 plants per population is required in order to make precise conclusions on population variables.
Response: We appreciate the reviewer’s concern regarding the sample size per population and its implications for the analysis of phenotypic variability and differentiation. Below, we provide a detailed explanation of our rationale and limitations:
We initially aimed to collect 20 plants per population as a balance between statistical rigor and the logistical constraints of managing the study. Considering that five populations per species were analysed, with a total of five species, this resulted in 500 plants cultivated and maintained under experimental conditions. This number was the maximum manageable within the available resources and experimental field capacity.
The variation in the number of plants per population (14-20) arose from plant mortality before experimental analyses were conducted (2 years after collections). This mortality was due to the challenges of plant establishment in controlled experimental conditions. Despite this reduction, we ensured that the remaining samples were representative of the populations.
We clarified this in the M&M section with the following sentence:
“Only plants that survived two years post-transplant in the field were used in the phenotypic and genetic analyses (N, Table 5).”
While we acknowledge that a minimum of 30 plants per population would enhance precision, our choice was guided by the trade-off between the scope of the study (which included five species and multiple populations) and resource limitations. Additionally, previous studies on Paspalum species with similar or slightly lower sample sizes have yielded meaningful and reproducible results regarding phenotypic and genetic diversity (see for example: Brugnoli et al., 2013; 2014; Sartor et al., 2013; Karunarathne & Hojsgaard, 2021; Reutemann et al. 2024).
Although the sample size may introduce some limitations, we used robust statistical methods and multiple populations per species to mitigate this effect and ensure reliable conclusions. The phenotypic and genetic patterns observed across populations and species strongly support the validity of our findings.
We appreciate the reviewer’s valuable feedback, and we incorporated these considerations into the discussion to acknowledge this limitation and provide context for our results:
“The low levels of phenotypic variation observed may also reflect a low representativeness of the sample sizes analysed for each population (N=14-20) and the constrains in detecting significant differentiation among individuals or populations [Hoban & Schlarbaum, 2014].”
Hoban, S., & Schlarbaum, S. (2014). Optimal sampling of seeds from plant populations for ex-situ conservation of genetic biodiversity, considering realistic population structure. Biological Conservation, 177, 90-99.
Brugnoli et al. Diversity in diploid, tetraploid, and mixed dip-loid-tetraploid populations of Paspalum simplex. Crop Sci. 2013, 53(4), 1509-1516.
Brugnoli et al. Diversity in apomictic populations of Paspalum simplex Morong. Crop Sci. 2014, 54, 1656-1664.
Sartor et al. Patterns of genetic diversity in natural populations of Paspalum agamic complexes. Plant Syst. Evol. 2013, 299, 1295-1306
Karunarathne, P.; Hojsgaard, D. Single independent autopolyploidization events from distinct diploid gene pools and residual sexuality support range expansion of locally adapted tetraploid genotypes in a South American grass. Front. Genet. 2021, 12, 736088
Reutemann et al. Comparative analysis of molecular and morphological diversity in two diploid Paspalum species (Poaceae) with contrasting mating systems. Plant Reprod. 2024, 37(1), 15-32
- There are some grammatical errors throughout, so please make sure you double check your manuscript for these.
Response: We tried to correct most of these errors.

Reviewer 3 Report
Comments and Suggestions for Authors
Revised Review Comments:
1. Supplement the Basic Description of the Paspalum Genus
The introduction to the Paspalum genus is somewhat brief and lacks comprehensive background information. It is recommended to include the following details: the total number of species within the Paspalum genus, and the number or proportion of species exhibiting self-fertilization, cross-fertilization, and apomixis. This additional information will help readers understand the distribution of the studied species within the genus and the significance of the research.
2. Support the Representativeness of the Study Species
The paper mentions that the five Paspalum species studied are common in the Argentine grasslands, but there is insufficient data to support their representativeness. It is recommended to include the following:
The ecological proportion of these species in grassland plant communities (e.g., their coverage or biomass contribution).
Their proportion within the Paspalum genus (e.g., population size or contribution to species diversity).
These additions will strengthen the argument for the representativeness of the chosen species in relation to the grassland or the genus.
3. Rationale for Sampling Sites
The paper does not provide sufficient details on the rationale for selecting the specific sampling sites or their representativeness. It is recommended to include:
The ecological and geographical characteristics of each sampling site.
The specific reasons for choosing these sites, such as whether they represent different ecological zones or habitat types.
Whether the selected sampling sites reflect the ecological and geographical diversity of the study region.
This additional information will further enhance the generalizability and scientific credibility of the results.
4. Interpretation of Results in Light of Existing Consensus
The paper finds minimal genetic differences between cross-fertilizing and self-fertilizing Paspalum species, which contradicts existing literature. It is advised that the authors:
Interpret these results with greater caution, avoiding overgeneralization of the findings.
Reassess the appropriateness of the selected study materials and experimental design (e.g., sample size, comprehensiveness of sampling).
Explore potential reasons for the unexpected findings (e.g., ecological adaptability, historical factors, gene flow), and discuss these in greater depth within the paper.
Author Response
We sincerely thank you for your valuable comments and constructive suggestions. We deeply appreciate the time and effort you dedicated to evaluating our manuscript. Your feedback has significantly contributed to improving the clarity, depth, and overall quality of our work. We have carefully addressed all your comments and, where applicable, provided a detailed rationale for our decisions. Your insightful critiques have greatly enhanced the scientific rigor of our manuscript, and we are truly grateful for the opportunity to incorporate these meaningful improvements.
Below we summarize responses to each specific point raised:
Reviewer: 1. Supplement the Basic Description of the Paspalum Genus
The introduction to the Paspalum genus is somewhat brief and lacks comprehensive background information. It is recommended to include the following details: the total number of species within the Paspalum genus, and the number or proportion of species exhibiting self-fertilization, cross-fertilization, and apomixis. This additional information will help readers understand the distribution of the studied species within the genus and the significance of the research.
Response: Thanks for this suggestion, we added the number of total species in the genus and the ones described as allogamous or apomictic. The number of species with selfing were already included in the introduction. We also included the number of polyploids studied.
The new paragraph says:
“Over 330 species have been described for this genus [Zuloaga et al. 2003], however only a few of them have been studied at a population level. […], 43.1% were self-sterile, and 75% were polyploids. Nevertheless, these species only represent 21.8% of the total number of species in the genus, and most studies involve singular plants rather than populations.”
Zuloaga, Fernando O., Morrone, Osvaldo, Davidse, Gerrit, Filgueiras, Tarciso S., Peterson, Paul M., Soreng, Robert J., and Judziewicz, Emmet J. 2003. "Catalogue of New World grasses (Poaceae): III. subfamilies Panicoideae, Aristidoideae, Arundinoideae, and Danthonioideae." Contributions from the United States National Herbarium. 46:1–662.
Reviewer: 2. Support the Representativeness of the Study Species
The paper mentions that the five Paspalum species studied are common in the Argentine grasslands, but there is insufficient data to support their representativeness. It is recommended to include the following:
- The ecological proportion of these species in grassland plant communities (e.g., their coverage or biomass contribution).
- Their proportion within the Paspalum genus (e.g., population size or contribution to species diversity).
These additions will strengthen the argument for the representativeness of the chosen species in relation to the grassland or the genus.
Response: Thanks for this suggestion, however, we could not find any report regarding the coverage proportion or biomass contribution in natural grasslands of the Paspalum species in this work. Also, most studies involving the genus focused on productive landscapes and forage species (e.g. Paspalum notatum, P. dilatatum, P. plicatulum), leading to biased results regarding coverage/dry biomass production of Paspalum species in natural backgrounds.
Nevertheless, most floristic studies in South American native grasslands reported a high contribution of Paspalum to species richness, and some species are usually reported as dominant (see for example: Longhi-Wagner et al. 2012; Pinto et al., 2013; Menezes et al., 2015; Andrade et al.2018; Oyarzabal et al., 2018).
Pallares et al (2005) reported that in Argentina, dry matter yield in natural grasslands has three main contributions: tussock prairies (pajonales, common species: Andropogon lateralis, Sorghastrum agrostoides, Paspalum quadrifarium and P. intermedium) with 5,077 kg DM/ha/year, short-grass grasslands (common species: Paspalum notatum, Axonopus argentinus, Sporobolus indicus) with 5,803 kg DM/ ha/year, and deteriorated grasslands (flechillares, common species: Aristida venustula) with 2,774 kg DM/ha/year. However, they did not report a DM yield for each singular dominant species.
Regarding the population size of the selected species, we selected populations that were 50 km away from each other, which sizes were ≥ 200 m (see Schedler et al. 2023 for collection site selection). In each population, we selected plants separated 10 m from each other (to exclude sampling genets/ramets). If we considered that each plant has a 1 m diameter (0.5-1.1m, data not shown), we can estimate that selected populations were N≈100-200 individuals.
Longhi-Wagner et al. (2012). Floristic affinities in montane grasslands in eastern Brazil. Systematics and Biodiversity, 10(4), 537-550.
Pinto et al. (2013). Floristic and vegetation structure of a grassland plant community on shallow basalt in southern Brazil. Acta Botanica Brasilica, 27, 162-179.
Menezes et al. (2015). Floristic and structural patterns in South Brazilian coastal grasslands. Anais da Academia Brasileira de Ciências, 87(4), 2081-2090.
Oyarzabal et al. (2018). Unidades de vegetación de la Argentina. Ecología austral, 28(1), 40-63.
Andrade et al (2018). Vascular plant species richness and distribution in the Río de la Plata grasslands. Botanical Journal of the Linnean Society, 188(3), 250-256.
Pallarés et al. (2005) Chapter 5: The South American Campos ecosystem. In Grasslands of the World. Edited by J.M. Suttie, S.G. Reynolds and C. Batello, pp. 171-219.
Reviewer: 3. Rationale for Sampling Sites
The paper does not provide sufficient details on the rationale for selecting the specific sampling sites or their representativeness. It is recommended to include:
- The ecological and geographical characteristics of each sampling site.
- The specific reasons for choosing these sites, such as whether they represent different ecological zones or habitat types.
- Whether the selected sampling sites reflect the ecological and geographical diversity of the study region.
This additional information will further enhance the generalizability and scientific credibility of the results.
Response: The sampling sites were selected based on:
- Natural distribution of the species in north-eastern Argentina: This include for ionanthum: Misiones & Corrientes; P. durifolium: Corrientes & Santa Fe; P. regnelli: Corrientes, Entre Rios & Misiones; P. urvillei: Chaco, Corrientes, Entre Ríos, Formosa, Misiones & Santa Fe; P. intermedium: Corrientes, Entre Ríos, Misiones, Santa Fe, Chaco & Formosa (Zuloaga & Morrone, 2005).
- Habitat sampling: from monotypic natural grasslands to transitional habitats affected by anthropogenic activities such as grazing or agriculture were taken into account. In natural grasslands, for example, populations of durifolium and P. intermedium were sampled in lowlands with periodic flooding, while P. ionanthum, P. regnellii and P. urvillei were collected from upland grasslands with well-drained soils. In disturbed habitat, for example roadside verges or pine plantation margins, we only found populations for P. regnellii and P. urvillei.
- Population sizes: We sampled populations with a lineal size ≥ 200m that were at least 50 km apart. Within populations we sampled individuals with a 10 m separation. Thus, if we considered a maximum plant diameter of 1m, each population could have a range of 100-200 individuals. Smaller populations smaller or with a lower density were excluded from this work.
- Ploidy composition of each population: We also considered, and only those that were monoploid and tetraploid were studied (see Schedler et al. 2023). Collected populations with mixed ploidy were also excluded from this work.
Thus, the resulting number and distribution of populations covered a latitudinal gradient and varying environmental conditions (e.g. soil type, hydrology, and vegetation structure.), population size and density, and ploidy composition.
Schedler et al. Alternative evolutionary pathways in Paspalum involving allotetraploidy, sexuality, and varied mating systems. Genes 2023, 14(6), 1137.
Zuloaga FO & Morrone O. Revisión de las especies de Paspalum para América del Sur austral: (Argentina, Bolivia, sur del Brasil, Chile, Paraguay y Uruguay). St. Louis, Missouri, USA: Missouri Botanical Garden, 2005
We included a short explanation in M&M section:
“Natural distribution of each species according to Zuloaga & Morrone (2005) was considered for population sampling. […] Within the natural area of each species, only populations with a lineal size ≥ 200 m that were monoploid and tetraploid were considered for this work (see Schedler et al. [47]). [..] Besides, habitat type was recorded for each species as undisturbed habitats (monotypic or transition grassland), and disturbed habitat (when the species was on road edges or forestation margins) (see Table 5).”
Zuloaga, F. O., & Morrone, O. (2005). Revisión de las Especies de Paspalum para América del Sur Austral (Argentina, Bolivia, sur de Brasil, Chile, Paraguay y Uruguay); Misouri Botanical Garden Press: St. Louis, MO, USA, 298.
Reviewer: 4. Interpretation of Results in Light of Existing Consensus
The paper finds minimal genetic differences between cross-fertilizing and self-fertilizing Paspalum species, which contradicts existing literature. It is advised that the authors:
- Interpret these results with greater caution, avoiding overgeneralization of the findings.
- Reassess the appropriateness of the selected study materials and experimental design (e.g., sample size, comprehensiveness of sampling).
- Explore potential reasons for the unexpected findings (e.g., ecological adaptability, historical factors, gene flow), and discuss these in greater depth within the paper.
Response: We appreciate the reviewer’s valuable feedback on the interpretation of our results. Below, we outline how we will address these points to strengthen the discussion and provide a more cautious interpretation of the findings.
Several factors may account for this unexpected result:
- The studied species inhabit overlapping environments with similar selective pressures, which may lead to convergent genetic patterns despite differing reproductive modes.
- Geographical barriers or anthropogenic habitat alterations may have restricted pollen and seed dispersal in cross-fertilizing species, reducing their genetic diversity.
- The sample size and representativeness of populations may have limited our ability to detect subtle genetic differences.
- Shared evolutionary histories among the studied species could explain the reduced genetic differentiation observed.
Thus, we included the following sentence in the conclusion section:
“While these findings provide valuable insights, future studies with larger sample sizes, broader geographical scopes, and additional molecular markers are necessary to validate these patterns and better understand the interplay between reproductive modes and genetic diversity in polyploid species of Paspalum.”

Round 2
Reviewer 2 Report
Comments and Suggestions for Authors
The authors have satisfactorily improved their manuscript according to my comments and they reported the limitations of their research study.
I believe that the paper can now be published.